# Patch Ranking Map: Explaining Relations among Top-Ranked Patches, Top-Ranked Features and Decisions of Convolutional Neural Networks for Image Classification

## Abstract

Since a conventional Convolutional Neural Network (CNN) using a large number of extracted features is not fully explainable and not very memory-efficient, we develop an explainable and efficient CNN model consisting of convolutional layers, a new feature selection (FS) layer, a classifier, and a novel "Patch Ranking Map" (PRM). The PRM contains top-ranked image patches that have important associations with decisions of the CNN. Top-ranked common features selected by different FS methods are used to generate two newly defined matrices: the "feature accumulation matrix" and the "feature ranking matrix". Different from a typical heatmap, these two matrices are used to rank image patches in the PRM to effectively explain the relationship among an input image, top-ranked features, top-ranked feature maps, and the final classification decision. Simulation results using the Alzheimer's MRI preprocessed dataset for 4-class image classification with $6,400$ $128 \times 128$ images indicate that the three PRMs based on three robust top-ranked common feature sets generated by seven different FS methods have the same top two most important patches associated with Alzheimer's disease diagnosis. In addition, $8 \times 8$ patches of a $128 \times 128$ image at the 7th and 12th patch rows and at the 8th and 9th patch columns are most informative because they have the top two most important patches and the top two most top-ranked common row-wise and column-wise features. The relationship among brain regions associated with Alzheimer's disease, the top-ranked patches, the top patch rows, and the top patch columns will be analyzed based on research results in brain informatics and medical informatics. The simulations also show that the trained CNN with FS can have higher classification accuracy and smaller model size than the conventional CNN without FS. More effective and efficient optimization algorithms will be developed to select the top (most informative) features and rank patches for building an accurate and efficient CNN model with more explainable decisions that can be captured by the PRM for various image classification applications.

## 1 Introduction

The Convolutional Neural Network (CNN) (Krizhevsky et al., 2012; LeCun et al., 2015) has successful applications for image classification, such as skin cancer detection (Esteva et al., 2012) and image recognition (He et al., 2016), but it is difficult to explain the reasons behind the black box model decisions. Being able to accurately interpret the decisions of CNN models is still an important problem, especially in computer vision.

Various methods are developed to interpret the decisions of a CNN (Wang et al., 2022; Schttl, 2022; Zhang et al., 2020; Kim et al., 2022). The class activation mapping (CAM) method using the global average pooling is developed to interpret a CNN by localizing the discriminative image regions (Zhou et al., 2016). The Grad-CAM applies gradients to generate heatmaps for visual explanations of a deep neural network (Selvaraju et al., 2017). A review of the latest applications of explainable deep learning methods for medical imaging discusses various approaches, challenges for clinical deployment, and the areas requiring further research (Singh et al., 2020). A detailed survey on

the explainability of machine learning algorithms presents different interpretations for distinct categories for the medical field (Tjoa & Guan, 2020). However, current methods such as CAM-methods based heatmaps do not deeply analyze a relationship among ranked image patches, top features, top feature maps, and a decision.

Feature selection (FS) is useful in not only improving the model performance but also in interpreting a deep neural network. MediMLP with FS using Grad-CAM was developed for lung cancer postoperative complication prediction (He et al., 2019). A new explainable vision transformer model was developed to find the most causally significant input regions that contribute to the model decision by using an instance-wise causal FS (Khanal et al., 2022). Open problems include how to select the best individual features from the feature maps to effectively improve the quality of visual explanations of a deep learning model, and how to use useful information in a small number of selected feature maps to deeply explain the relationship among an input image, selected features, and a final decision.

Research works in identifying the top image regions associated with Alzheimer's disease have been done in recent years. For example, an expert physician identified seven regions of interest in the fluorodeoxyglucose positron emission tomography images related to the Alzheimer's disease (Aidos et al.). The assessment of shape-based analysis of key brain regions indicated that the scale-invariant heat kernel signature was predictive of disease stage and disease progression (Duarte et al.). A significant research problem is how to develop a novel method to effectively explain the relationship among ranked top input image patches associated with special regions such as brain regions, ranked top features, and the final decisions of the CNN.

In this paper, we propose an efficient and accurate CNN image classification model by adding a novel layer called the "FS Layer." The FS layer uses multiple FS methods to reliably select the top image features that are used to build an accurate and efficient CNN model. New informative feature structures such as top feature maps, feature accumulation matrix, the feature distribution matrix, and the top patch map are newly defined to effectively explain the relationship among an input image, top features, top feature maps, top patch maps and a final decision.

## 2 NEW INFORMATIVE FEATURE STRUCTURES

### 2.1 TOP FEATURE MAP

The last Maxpooling layer of a CNN generates $n$ $H \times W$ feature maps $F^l$ (assuming the feature map shape is $H \times W \times n$) having features $f_{ij}^l$ for $i = 0, 1, \ldots, H - 1$, $j = 0, 1, \ldots, W - 1$, and $l = 0, 1, \ldots, n - 1$. The $n$ feature maps are converted to $m$ flattened features for $m = n \times H \times W$. The $m$ features have $m$ relevant feature index numbers (0, 1, …, $m - 1$). A FS method selects top $k$ features from the $m$ features. The $k$ selected features have $k$ feature index numbers $I_p$ for $I_p \in 0, 1, \ldots, m - 1$ for $p = 0, 1, \ldots, k - 1$. A top feature with $I_p$ is associated with a feature map $F^{q_p}$ where $q_p = I_p \bmod n$ for $p = 0, 1, \ldots, k - 1$. Let $\bar{Q} = \{q_0, q_1, \ldots, q_{k-1}\}$. After eliminating duplicated elements in $Q_S$, we get $Q$ with distinct elements for $Q \subseteq \bar{Q}$.

**Definition 1**: Let the top feature map $T^q$ have features $t_{ij}^q$ for $i = 0, 1, \ldots, H-1, j = 0, 1, \ldots, W - 1$, and $q \in Q$. If $f_{ij}^q$ in a feature map $F^q$ is a selected feature, then $t_{ij}^q = f_{ij}^q$, otherwise $t_{ij}^q = 0$.

The top feature map selection method has two steps: (1) identifying feature maps $F^q$ for $q \in Q$, and (2) generating relevant top feature maps $T^q$ for $q \in Q$ based on Definition 1.

Let $P$ be the number of top feature maps (i.e., the number of elements in $Q$), so we have $\lceil k/(H \times W) \rceil \leq P \leq n$. In addition, the distribution of a small number of top feature maps among $n$ feature maps $F^l$ for $l = 0, 1, \ldots, n - 1$ is useful to understand the relationship among an input image, selected features, and the final decision.

### 2.2 INFORMATIVE FEATURE MATRICES

To sufficiently investigate the relationship among an input image, the top features, the top feature maps, and the final decision, we define six new feature structures as follows.

**Definition 2**: Let the "feature binary matrix" $B$ have binary numbers $b_{ij}$ for $i = 0, 1, \ldots, H - 1$, and $j = 0, 1, \ldots, W - 1$. If $f_{ij}$ is a selected feature, then $b_{ij} = 1$, otherwise $b_{ij} = 0$.

For a special case, for $n$ feature maps $T^l$ for $l = 0, 1, \ldots, n - 1$, the relevant feature binary matrices $B^l$ with $b_{ij}^l = 1$ for $i = 0, 1, \ldots, H - 1$, $j = 0, 1, \ldots, W - 1$, and $l = 0, 1, \ldots, n - 1$ because FS is not performed.

**Definition 3**: Let the "feature accumulation matrix" $A$ have elements called "feature accumulators" $a_{ij}$ for $i = 0, 1, \ldots, H - 1$ and $j = 0, 1, \ldots, W - 1$, where $a_{ij} = \sum_{s=1}^{K} b_{ij}^s$ where $b_{ij}^s$ is an element of the feature binary matrix $B^s$, and $K$ is the number of feature maps $T$.

**Definition 4**: The row-wise feature accumulation number $A_{row}^i$ is defined as $A_{row}^i = \sum_{j=0}^{W-1} a_{ij}$ for $i = 0, 1, \ldots, H - 1$.

**Definition 5**: The column-wise feature accumulation number $A_{column}^j$ is defined as $A_{column}^j = \sum_{i=0}^{H-1} a_{ij}$ for $j = 0, 1, \ldots, W - 1$.

**Definition 6**: Let the "feature distribution matrix" $D$ have elements $d_{ij}$ for $i = 0, 1, \ldots, H - 1$ and $j = 0, 1, \ldots, W - 1$, where $d_{ij} = a_{ij}/k$ where $a_{ij}$ are feature accumulators of the feature accumulation matrix $A$.

Let the top feature map $T^q$ have features $t_{ij}^q$ for $i = 0, 1, \ldots, H - 1$, $j = 0, 1, \ldots, W - 1$, and $q \in Q$. If $f_{ij}^q$ in a feature map $F^q$ is a selected feature, then $t_{ij}^q = f_{ij}^q$, otherwise $t_{ij}^q = 0$. The features $t_{ij}^q$ are ranked by a feature ranking method such as the RFE (Guyon et al., 2002), then each feature has its ranking number $r_{ij}^q$ for $i = 0, 1, \ldots, H - 1$, $j = 0, 1, \ldots, W - 1$, and $q \in Q$, where the lower a ranking number, the higher a feature ranking. $r_{ij}^q$ are sorted to generate new ranking numbers $\bar{r}_{ij}^k$ in an increasing order for $k = 0, 1, \ldots, a_{ij} - 1$.

**Definition 7**: Let the "feature ranking matrix" $R^k$ have ranking numbers $\bar{r}_{ij}^k$ where $\bar{r}_{ij}^k \leq \bar{r}_{ij}^{k+1}$ for $i = 0, 1, \ldots, H - 1$, $j = 0, 1, \ldots, W - 1$, and $k = 0, 1, \ldots, a_{ij} - 1$, where $a_{ij}$ are feature accumulators of the feature accumulation matrix $A$.

### 2.3 THE PATCH RANKING MAP

To better understand the relationship between input images and final decisions, it is useful to rank image patches with degrees of importance for decisions. A new definition is given below.

**Definition 8**: Let the "patch ranking matrix" $P$ have patch ranking numbers $p_{ij}$ for $p_{ij} \in \{1, 2, \ldots, HW\}$ for $i = 0, 1, \ldots, H - 1$, $j = 0, 1, \ldots, W - 1$. The smaller $p_{ij}$ is, the more important a patch at $(i, j)$ is associated with a decision.

## 3 MULTI-PHASE FS ALGORITHMS

We propose a general multi-phase FS algorithm, as shown in Algorithm 1. Different FS methods at different phases are used to find a small set of robust and effective features based on different selection criteria. Thus, Algorithm 1 with multiple phases can reliably select top-ranked features by different FS methods.

---

**Algorithm 1** General $m$-Phase FS Algorithm for $m > 1$

---

**Input:** Training Dataset and Test Dataset with $N$ Features
**Output:** Training Dataset and Test Dataset with $K$ Features
  1: Phase 1: Apply the 1st FS Method to Generate a New Training Dataset and a New Test Dataset with $N_1$ Selected Features for $N_1 < N$.
  2: Phase 2: Apply the 2nd FS method to Generate a New Training Dataset and a New Test Dataset with $N_2$ Features for $N_2 < N_1$.
  3: ...
  4: Phase $m$: Apply the $m$-th FS method to Generate a Final Training Dataset and a Final Test Dataset with $K$ Features for $K < N_{m-1}$.

---

## 4 THE ALGORITHM FOR RANKING PATCHES

The explainable and efficient CNN framework, as shown in Fig. 1, consists of the following: (1) convolutional layers for extracting $n$ $H \times W$ feature maps from a $\bar{H} \times \bar{W}$ input image, (2) the flatten layer for converting $m$ flattened features from the $n$ $H \times W$ feature maps for $m = n \times H \times W$, (3) the FS layer for selecting top $k$ features from the $m$ flattened features, (4) a patch ranking method for generating a PRM using the top $k$ features, and (5) a classifier for making a decision based on the top $k$ features. Since the $H \times W$ feature map with the top $k$ features are associated with $\bar{H} \times \bar{W}$ input image with $\bar{H}/H \times \bar{W}/W$ patches ($\bar{H}$ is divisible by $H$ and $\bar{W}$ is divisible by $W$ for even patch distribution for the PRM), the top $k$ features rankings related to decisions can be used to rank relevant image patches. The ranked image patches are useful for a user, such as a medical doctor, to understand which image regions are most important to help with making a final diagnosis decision.

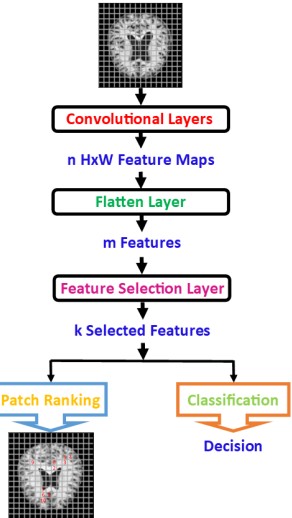

Figure 1: Explainable and Efficient CNN Model with a Top Patch Map.

The higher value of an element at a location $(i, j)$ of the feature accumulation matrix and the higher ranking of a top feature at a location $(i, j)$ in the feature ranking matrix for $i = 0, 1, \ldots, H - 1$ and $j = 0, 1, \ldots, W - 1$, the more important a patch at a location $(i, j)$. We propose a new image patch ranking algorithm based on both a feature accumulation matrix and a feature ranking matrix $R^0$ with the top features with their ranking numbers $\bar{r}_{ij}^0$. It is shown in Algorithm 2.

---

**Algorithm 2** The Algorithm for Ranking Patches

**Input:** A feature accumulation matrix $A$ and a feature ranking matrix $R^0$ with top features' ranking scores $\bar{r}_{ij}^0$.
**Output:** $K$ top patches.
1: Step 1: Calculate a ranking score of a patch at $(i, j)$: $\theta_{ij} = f(a_{ij}, \bar{r}_{ij}^0)$.
2: Step 2: Sort patch ranking scores.
3: Step 3: Select top $K$ patches.

---

A feature ranking score is defined as $\beta_{ij} = 1 - (\bar{r}_{ij}^0 - 1)/(n - 1)$ where $n$ is the total number of top features, and a feature accumulation ranking score is defined as $\alpha_{ij} = 1 - (a_{ij} - 1)/(m - 1)$ for $m = max(a_{ij})$ for $i = 0, 1, \ldots, H-1$, and $j = 0, 1, \ldots, W-1$. To get a robust patch ranking score, different weighted average functions for ranking patches are defined as $\theta_{ij}^k = \omega_k \beta_{ij} + (1 - \omega_k)\alpha_{ij}$ where $\omega_{k+1} = \omega_k + \delta$, $0 < \omega_k < 1$ and $0 < \delta < 1$ for $k = 1, 2, \ldots, K$ and $K$ is an odd positive integer. Then the reliable patch ranking function is defined as an average of the different weighted average functions: $\theta_{ij} = (\sum_{k=1}^{K} \theta_{ij}^k)/K$ for $0 \le \theta_{ij} \le 1$. Thus, $\theta_{ij} = (\omega_1 + \delta(K - 1)/2)(\beta_{ij} - 1)$

$\alpha_{ij}) + \alpha_{ij} = \omega_{(K-1)/2}(\beta_{ij} - \alpha_{ij}) + \alpha_{ij}$. For a given interval [a, b] of $\omega$, a patch ranking function can be defined as $\theta_{ij} = \bar{\omega}(\beta_{ij} - \alpha_{ij}) + \alpha_{ij}$ for $\bar{\omega} = (a + b)/2$.

## 5 PERFORMANCE ANALYSIS

The Alzheimer's MRI preprocessed dataset with $6,400$ $128 \times 128$ images (Kumar & Shastri, 2022) are used for performance analysis. It has 4 classes: Mild Demented (896 images), Moderate Demented (64 images), Non Demented ($3,200$ images), and Very Mild Demented ($2,240$ images). The $6,400$ images are divided into $5,119$ training images, 642 test images, and 639 validation images.

We used a feature extractor that consists of convolutional layers of a ResNet50 model (pretrained on ImageNet dataset and fine-tuned using the Alzheimer's MRI dataset) and a new MaxPooling layer. The final MaxPooling layer of the fine-tuned pretrained ResNet50 generates 64 $16 \times 16$ feature maps. The $16 \times 16$ feature map has 256 features that are associated with 256 $8 \times 8$ patches. The $16,384$ features have feature index numbers (i.e., 0, 1, ..., $16,383$).

### 5.1 FS METHODS FOR BUILDING ACCURATE AND MEMORY-EFFICIENT CLASSIFIER

We use seven different FS methods, including three 1-phase FS algorithms using Chi2, f_regression and f_classif respectively, the 2-phase FS algorithm using f_regression and the RFE, the 3-phase FS algorithm using Chi2, f_classif and the RFE, the 4-phase FS algorithm using Chi2, mutual_info_regression, f_classif and the RFE, and the 5-phase FS algorithm using Chi2, f_regression, mututal_info_regression, f_classif and the RFE. These seven different FS methods are used to generate seven different top feature sets from all $16,384$ features. The RFE method is applied to rank top features for each top feature set. The Alzheimer's MRI data with a small number of top features are used to train a MLP model. Finally, the ResNet50-FS model is built by combining the fine-tuned pretrained ResNet50 model without the MLP, the new FS layer, and the newly trained MLP model.

Tables 1 and 2 show that the trained ResNet50-FS models can have higher classification accuracy and smaller model size than the conventional ResNet50 without FS with test accuracy 0.9642. The top features are selected by the 5-phase FS algorithm from $16,384$ features with original 64 $16 \times 16$ feature maps. Top feature maps associated with selected features, test accuracies and model sizes are shown in Table 1. The traditional ResNet50 without FS uses 64 256-feature maps, the new ResNet50-FS model uses fewer top feature maps with much smaller number of selected top features. For instance, a ResNet50-FS model with the top 100 features uses the top 41 feature maps (23 original feature maps and $16,284$ features are eliminated from 64 original feature maps). The top 41 feature maps index numbers include 1, 3, 4, 6, 7, 8, 10, 11, 13, 14, 15, 18, 19, 20, 21, 23, 24, 26, 27, 28, 30, 32, 33, 35, 37, 38, 39, 41, 43, 44, 45, 48, 49, 51, 53, 54, 57, 58, 59, 60, and 61. For example, after 251 features of the 27th original feature map with an index number 26 are eliminated by the 5-phase FS method, a top feature map with top five features ($F_1$=1306, $F_2$=5786, $F_3$=6490, $F_3$=6746, and $F_5$=11738) is generated as shown in Fig. 8 in Appendix.

Table 1: Performance of a Model with All Features and Efficient Models with Top Features

| Number of Features | $16,384$ | $1,000$ | 800 | 600 | 400 | 200 | 100 |
|---|---|---|---|---|---|---|---|
| Number of Feature Maps | 64 | 64 | 63 | 63 | 63 | 57 | 41 |
| Test Accuracy | 0.9642 | 0.9844 | 0.9891 | 0.9688 | 0.9657 | 0.9299 | 0.8738 |
| Model Size (KB) | $16,300$ | 696 | 594 | 268 | 241 | 184 | 156 |

Table 2: Test Accuracies of a Model with $16,384$ Features and Models with $800$ Selected Features

| No FS | Chi2 | f_classif | f_regression | 2-phase FS | 3-phase FS | 4-phase FS | 5-phase FS |
|---|---|---|---|---|---|---|---|
| 0.9642 | 0.9798 | 0.9704 | 0.9611 | 0.9657 | 0.9704 | 0.9704 | 0.9891 |

## 5.2 Multiple FS Methods for Selecting Reliable Top Common Features

The top seven partial feature sets with top six features from the complete top seven 100-feature sets generated the seven FS methods are shown in Table 3. The feature (11738) is ranked first for all top seven 100-feature sets, so it is the most important feature. The second most important feature is 6693. Importantly, 11738 is also ranked first in both the top seven 400-feature sets and the top seven 800-feature sets, so it is the most important feature associated with the patch at (11, 7) for Alzheimer's disease diagnosis.

To get reliable top common features, the top 188, 65, and 12 common features are selected based on average feature rankings from seven 800-feature sets, seven 400-feature sets, and seven 100-feature sets generated by the seven FS methods.

Table 3: Seven Top-ranked 6 Feature Sets Generated by the Seven FS Methods

| Chi2 | f_classif | f_regression | 2-phase FS | 3-phase FS | 4-phase FS | 5-phase FS |
|---|---|---|---|---|---|---|
| **11738** | **11738** | **11738** | **11738** | **11738** | **11738** | **11738** |
| 5655 | **6693** | 5638 | 5638 | **6693** | **6693** | **6693** |
| 4737 | 7437 | **6693** | 10073 | 5030 | 6664 | 5030 |
| 15002 | 6811 | 4776 | 15002 | 7437 | 7437 | 6664 |
| 1330 | 6684 | 10151 | **6693** | 5786 | 6811 | 7437 |
| **6693** | 6700 | 11649 | 15039 | 6700 | 5390 | 6811 |

## 5.3 Top Patch Maps Based on the Feature Accumulation Matrix and the Feature Ranking Matrix

The feature accumulation matrices and the top feature ranking matrices for the top 188, 65 and 12 common features are shown in Figures 2, 3, and 4, respectively. A top feature ranking matrix $R^0$ has $\bar{t}_{ij}^0$ for $i = 0, 1, \ldots, 15$ and $j = 0, 1, \ldots, 15$. The other top feature ranking matrices $R^k$ for $k = 1, 2, \ldots, a_{ij}$ have other ranking numbers. For instance, the other top three feature ranking matrices for the top 12 common features are shown in Figures 9, 10, and 11 in Appendix. The right column of the feature accumulation matrix shows blue row-wise feature accumulation numbers $A_{row}^i$ for $i = 0, 1, \ldots, 15$. The bottom row of the feature accumulation matrix shows green column-wise feature accumulation numbers $A_{column}^j$ for $j = 0, 1, \ldots, 15$.

Fig. 2 (a) shows that the top three patch row numbers are 6, 11, and 4. Fig. 3 (a) and Fig. 4 (a) show that the top three patch row numbers are 6, 11, and 5. Fig. 2 (a), Fig. 3 (a) and Fig. 4 (a) show that the top three patch column numbers are 7, 8, and 10. $8 \times 8$ patches of a $128 \times 128$ image at the top patch rows and the top patch columns are more informative for image classification because they have more top-ranked common features. Thus, the relationship between Alzheimer's disease and both the top patch rows and the top patch columns will be investigated.

For three intervals [0.2, 0.4], [0.4, 0.6], and [0.6, 0.8] of $\omega$, $\omega = 0.3$, $\omega = 0.5$, and $\omega = 0.7$ are used for the patch ranking function $\theta_{ij} = \omega \times (\beta_{ij} - \alpha_{ij}) + \alpha_{ij}$. The ranked top 10, 10, and 7 $8 \times 8$ patches are shown in three $16 \times 16$ PRMs in Fig. 5, Fig. 6, and Fig. 7(a), respectively.

Fig. 5, Fig. 6, and Fig. 7(a) show that the top two patches' rankings for $\omega = 0.3, 0.5$, and $0.7$ are unchanged for the top 188, 65, and 12 common features. Interestingly, the top three patches are located at the intersections of the top three rows and the top three columns. For example, the top three patches in Fig. 6 are located at (11, 7), (6, 8) and (5, 10) for the top three rows (6, 11, 5) and the top three columns (7, 8, 10). Fig. 7(a) shows that all top seven patches' rankings for $\omega = 0.3, 0.5$, and $0.7$ are unchanged. Fig. 7(b) and Fig. 7(c) indicate that the top two patches associated with the top two features (11738 and 6693) are most informative for Alzheimer's disease diagnosis.

The feature accumulation matrices in Figures 2, 3, and 4 show that less important features are gradually eliminated from both the four side black areas and the center butterfly black area. Also, they show that the more important top 400 features and the most important top 100 features are generated by ranking common features extracted from seven feature sets generated by the seven FS methods. Similarly, the top-ranked patches based on the top 100 features in Fig. 7(a) are more

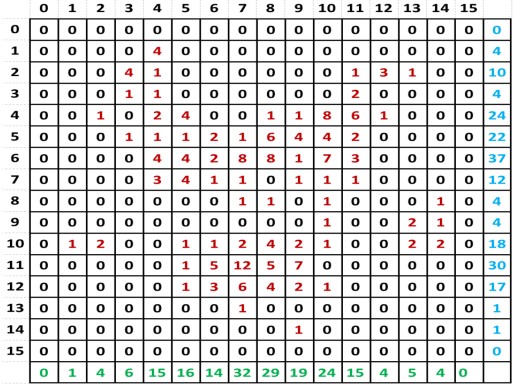

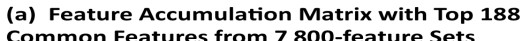

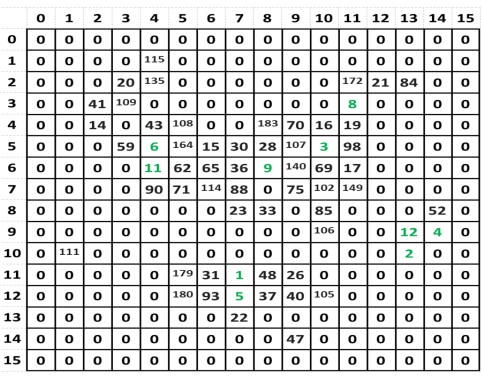

(a) Feature Accumulation Matrix with Top 188 Common Features from 7 800-feature Sets

(b) Feature Ranking Matrix with Top 10 Common Features in Green among 71 Ones

Figure 2: A Feature Accumulation Matrix and a Feature Ranking Matrix $R^0$ (Top 188 Common Features).

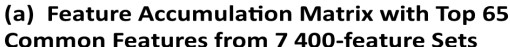

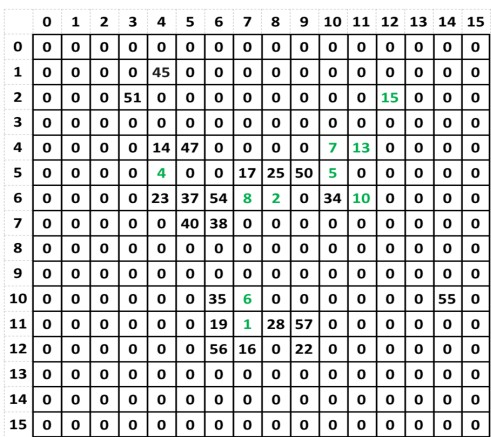

(a) Feature Accumulation Matrix with Top 65 Common Features from 7 400-feature Sets

(b) Feature Ranking Matrix with Top 10 Common Features in Green among 31 Ones

Figure 3: A Feature Accumulation Matrix and a Feature Ranking Matrix $R^0$ (Top 65 Common Features).

informative and more critical for decision-making than those based on the top 800 and 400 features in Figures 5 and 6. The most important patch located at (11, 7) in Figures 5, 6, and 7 is associated with both the largest number of features in the feature accumulation matrices in Figures 2(a), 3(a), and 4(a) and the most important feature (11738) in Figures 2(b), 3(b) and 4(b).

In summary, the relationship among the top seven patches in Fig. 7(a), the top two patch rows (6 and 11) and the two top patch columns (7 and 8) in Figures 2, 3 and 4 and Alzheimer's disease will be analyzed by using results in relevant literature.

## 6 CONCLUSIONS

Simulation results using the Alzheimer's MRI preprocessed dataset indicate that three $16 \times 16$ PRMs have reliable ranked top $8 \times 8$ patches for an explainable diagnosis. For instance, the three PRMs based on the top 188, 65, and 12 common features have the same top two most important patches

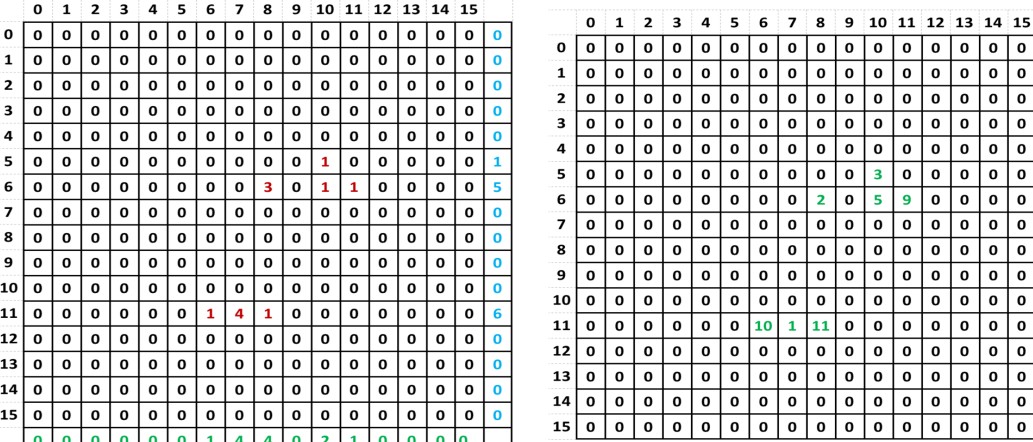

(a) Feature Accumulation Matrix with Top 12 Common Features from 7 100-feature Sets

(b) Feature Ranking Matrix with Top 7 Common Features in Green

Figure 4: A Feature Accumulation Matrix and a Feature Ranking Matrix $R^0$ (Top 12 Common Features).

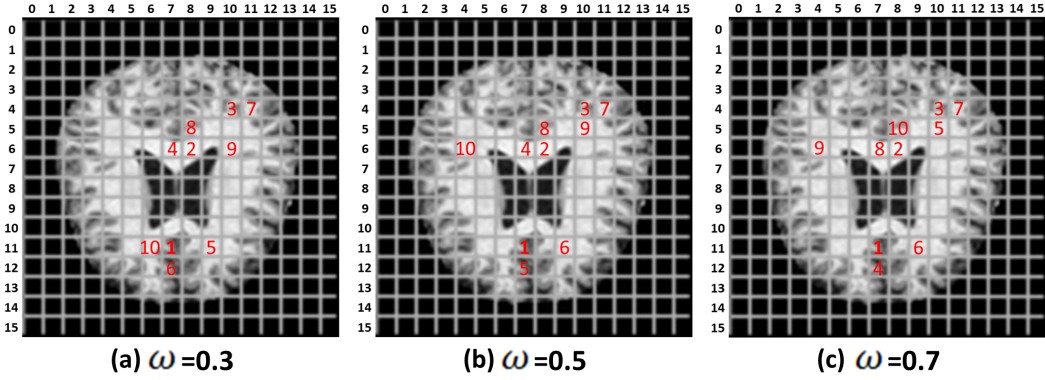

(a) $\omega$ =0.3      (b) $\omega$ =0.5      (c) $\omega$ =0.7

Figure 5: Top Three Patch Maps with Different Values of $\omega$ Based on the Top 188 Common Features.

associated with Alzheimer's disease diagnosis. In addition, $8 \times 8$ patches of a $128 \times 128$ image at the 7th and 12th patch rows and at the 8th and 9th patch columns are most informative because they have the top two most important patches and most top-ranked common features.

A new and efficient CNN model consists of convolutional layers, the new FS layer, a classifier, and the new PRM. The simulations also show that the trained CNN with FS can have higher classification accuracy and smaller model size than the conventional CNN without FS.

The explainable top-ranked image patches are associated with decisions of the CNN because the top patches are ranked based on highly informative feature accumulation matrix and feature ranking matrix. The PRM can provide users, such as medical doctors, with ranked patches based on a small number of highly informative features for them to better understand the relationship between the input images and the decisions of a CNN with FS.

A small number of informative top feature maps associated with the selected informative image features can be easily generated. The feature accumulation matrices, the feature distribution matrices, and feature ranking matrices have much statistical information associated with an input image, top patches, top features, top feature maps, and the final decision.

With the decrease of the top features selected by the FS method, gradual changes of the feature accumulation matrices, the feature distribution matrices, and PRMs can be used to analyze the rela-

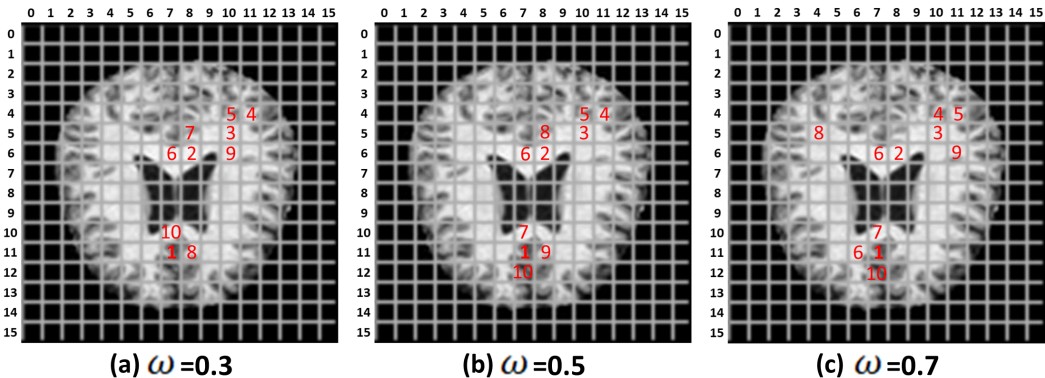

Figure 6: Top Three Patch Maps with Different Values of $\omega$ Based on the Top 65 Common Features.

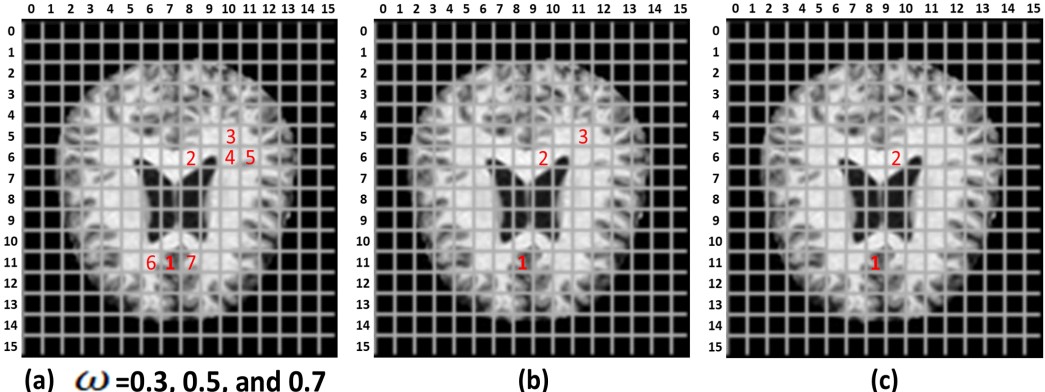

Figure 7: (a) The Top Patch Map Based on the Top 12 Common Features, (b) The Same Ranked Top Patches in Figures 6 and 7(a), (c) The Same Ranked Top Patches in Figures 5, 6, and 7(a).

tions among the most, less, and least important features and the most, less, and least important input image patches, and the final decisions of a CNN with FS.

# 7    FUTURE WORKS

It is useful and necessary to sufficiently analyze the relationship among an input image, image patches, selected top features, top feature maps, and the final decision in statistical and mathematical ways by using the feature accumulation matrices, feature distribution matrices, feature ranking matrices, PRMs, and heatmaps together.

The relationship among Alzheimer's disease, the top patches and both the top patch rows, and the top patch columns will be analyzed by using brain regions identified by experts and research results in brain informatics and medical informatics.

More effective and efficient optimization algorithms will be developed to select the top (most informative) features and rank patches for building an accurate and efficient CNN model with more explainable decisions that can be captured by the PRM for various image classification applications.

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

# A  APPENDIX

|    | 0 | 1 | 2 | 3 | 4 | 5 | 6 | 7 | 8 | 9 | 10 | 11 | 12 | 13 | 14 | 15 |
|----|---|---|---|---|---|---|---|---|---|---|----|----|----|----|----|----|
| 0  | 0 | 0 | 0 | 0 | 0 | 0 | 0 | 0 | 0 | 0 | 0 | 0 | 0 | 0 | 0 | 0 |
| 1  | 0 | 0 | 0 | 0 | $F_1$ | 0 | 0 | 0 | 0 | 0 | 0 | 0 | 0 | 0 | 0 | 0 |
| 2  | 0 | 0 | 0 | 0 | 0 | 0 | 0 | 0 | 0 | 0 | 0 | 0 | 0 | 0 | 0 | 0 |
| 3  | 0 | 0 | 0 | 0 | 0 | 0 | 0 | 0 | 0 | 0 | 0 | 0 | 0 | 0 | 0 | 0 |
| 4  | 0 | 0 | 0 | 0 | 0 | 0 | 0 | 0 | 0 | 0 | 0 | 0 | 0 | 0 | 0 | 0 |
| 5  | 0 | 0 | 0 | 0 | 0 | 0 | 0 | 0 | 0 | 0 | 0 | $F_2$ | 0 | 0 | 0 | 0 |
| 6  | 0 | 0 | 0 | 0 | 0 | $F_3$ | 0 | 0 | 0 | $F_4$ | 0 | 0 | 0 | 0 | 0 | 0 |
| 7  | 0 | 0 | 0 | 0 | 0 | 0 | 0 | 0 | 0 | 0 | 0 | 0 | 0 | 0 | 0 | 0 |
| 8  | 0 | 0 | 0 | 0 | 0 | 0 | 0 | 0 | 0 | 0 | 0 | 0 | 0 | 0 | 0 | 0 |
| 9  | 0 | 0 | 0 | 0 | 0 | 0 | 0 | 0 | 0 | 0 | 0 | 0 | 0 | 0 | 0 | 0 |
| 10 | 0 | 0 | 0 | 0 | 0 | 0 | 0 | 0 | 0 | 0 | 0 | 0 | 0 | 0 | 0 | 0 |
| 11 | 0 | 0 | 0 | 0 | 0 | 0 | 0 | $F_5$ | 0 | 0 | 0 | 0 | 0 | 0 | 0 | 0 |
| 12 | 0 | 0 | 0 | 0 | 0 | 0 | 0 | 0 | 0 | 0 | 0 | 0 | 0 | 0 | 0 | 0 |
| 13 | 0 | 0 | 0 | 0 | 0 | 0 | 0 | 0 | 0 | 0 | 0 | 0 | 0 | 0 | 0 | 0 |
| 14 | 0 | 0 | 0 | 0 | 0 | 0 | 0 | 0 | 0 | 0 | 0 | 0 | 0 | 0 | 0 | 0 |
| 15 | 0 | 0 | 0 | 0 | 0 | 0 | 0 | 0 | 0 | 0 | 0 | 0 | 0 | 0 | 0 | 0 |

Figure 8: The $16 \times 16$ Top Feature Map with top five features generated by the 5-phase FS method.

|    | 0 | 1 | 2 | 3 | 4 | 5 | 6 | 7 | 8 | 9 | 10 | 11 | 12 | 13 | 14 | 15 |
|----|---|---|---|---|---|---|---|---|---|---|----|----|----|----|----|----|
| 0  | 0 | 0 | 0 | 0 | 0 | 0 | 0 | 0 | 0 | 0 | 0 | 0 | 0 | 0 | 0 | 0 |
| 1  | 0 | 0 | 0 | 0 | 0 | 0 | 0 | 0 | 0 | 0 | 0 | 0 | 0 | 0 | 0 | 0 |
| 2  | 0 | 0 | 0 | 0 | 0 | 0 | 0 | 0 | 0 | 0 | 0 | 0 | 0 | 0 | 0 | 0 |
| 3  | 0 | 0 | 0 | 0 | 0 | 0 | 0 | 0 | 0 | 0 | 0 | 0 | 0 | 0 | 0 | 0 |
| 4  | 0 | 0 | 0 | 0 | 0 | 0 | 0 | 0 | 0 | 0 | 0 | 0 | 0 | 0 | 0 | 0 |
| 5  | 0 | 0 | 0 | 0 | 0 | 0 | 0 | 0 | 0 | 0 | 0 | 0 | 0 | 0 | 0 | 0 |
| 6  | 0 | 0 | 0 | 0 | 0 | 0 | 0 | 0 | 8 | 0 | 0 | 0 | 0 | 0 | 0 | 0 |
| 7  | 0 | 0 | 0 | 0 | 0 | 0 | 0 | 0 | 0 | 0 | 0 | 0 | 0 | 0 | 0 | 0 |
| 8  | 0 | 0 | 0 | 0 | 0 | 0 | 0 | 0 | 0 | 0 | 0 | 0 | 0 | 0 | 0 | 0 |
| 9  | 0 | 0 | 0 | 0 | 0 | 0 | 0 | 0 | 0 | 0 | 0 | 0 | 0 | 0 | 0 | 0 |
| 10 | 0 | 0 | 0 | 0 | 0 | 0 | 0 | 0 | 0 | 0 | 0 | 0 | 0 | 0 | 0 | 0 |
| 11 | 0 | 0 | 0 | 0 | 0 | 0 | 0 | 4 | 0 | 0 | 0 | 0 | 0 | 0 | 0 | 0 |
| 12 | 0 | 0 | 0 | 0 | 0 | 0 | 0 | 0 | 0 | 0 | 0 | 0 | 0 | 0 | 0 | 0 |
| 13 | 0 | 0 | 0 | 0 | 0 | 0 | 0 | 0 | 0 | 0 | 0 | 0 | 0 | 0 | 0 | 0 |
| 14 | 0 | 0 | 0 | 0 | 0 | 0 | 0 | 0 | 0 | 0 | 0 | 0 | 0 | 0 | 0 | 0 |
| 15 | 0 | 0 | 0 | 0 | 0 | 0 | 0 | 0 | 0 | 0 | 0 | 0 | 0 | 0 | 0 | 0 |

Figure 9: The Feature Ranking Matrix $R^1$ with the top two features from the top 12 common features.

|    | 0 | 1 | 2 | 3 | 4 | 5 | 6 | 7 | 8 | 9 | 10 | 11 | 12 | 13 | 14 | 15 |
|----|---|---|---|---|---|---|---|---|---|---|----|----|----|----|----|----|
| 0  | 0 | 0 | 0 | 0 | 0 | 0 | 0 | 0 | 0 | 0 | 0 | 0 | 0 | 0 | 0 | 0 |
| 1  | 0 | 0 | 0 | 0 | 0 | 0 | 0 | 0 | 0 | 0 | 0 | 0 | 0 | 0 | 0 | 0 |
| 2  | 0 | 0 | 0 | 0 | 0 | 0 | 0 | 0 | 0 | 0 | 0 | 0 | 0 | 0 | 0 | 0 |
| 3  | 0 | 0 | 0 | 0 | 0 | 0 | 0 | 0 | 0 | 0 | 0 | 0 | 0 | 0 | 0 | 0 |
| 4  | 0 | 0 | 0 | 0 | 0 | 0 | 0 | 0 | 0 | 0 | 0 | 0 | 0 | 0 | 0 | 0 |
| 5  | 0 | 0 | 0 | 0 | 0 | 0 | 0 | 0 | 0 | 0 | 0 | 0 | 0 | 0 | 0 | 0 |
| 6  | 0 | 0 | 0 | 0 | 0 | 0 | 0 | 0 | 12 | 0 | 0 | 0 | 0 | 0 | 0 | 0 |
| 7  | 0 | 0 | 0 | 0 | 0 | 0 | 0 | 0 | 0 | 0 | 0 | 0 | 0 | 0 | 0 | 0 |
| 8  | 0 | 0 | 0 | 0 | 0 | 0 | 0 | 0 | 0 | 0 | 0 | 0 | 0 | 0 | 0 | 0 |
| 9  | 0 | 0 | 0 | 0 | 0 | 0 | 0 | 0 | 0 | 0 | 0 | 0 | 0 | 0 | 0 | 0 |
| 10 | 0 | 0 | 0 | 0 | 0 | 0 | 0 | 0 | 0 | 0 | 0 | 0 | 0 | 0 | 0 | 0 |
| 11 | 0 | 0 | 0 | 0 | 0 | 0 | 0 | 6 | 0 | 0 | 0 | 0 | 0 | 0 | 0 | 0 |
| 12 | 0 | 0 | 0 | 0 | 0 | 0 | 0 | 0 | 0 | 0 | 0 | 0 | 0 | 0 | 0 | 0 |
| 13 | 0 | 0 | 0 | 0 | 0 | 0 | 0 | 0 | 0 | 0 | 0 | 0 | 0 | 0 | 0 | 0 |
| 14 | 0 | 0 | 0 | 0 | 0 | 0 | 0 | 0 | 0 | 0 | 0 | 0 | 0 | 0 | 0 | 0 |
| 15 | 0 | 0 | 0 | 0 | 0 | 0 | 0 | 0 | 0 | 0 | 0 | 0 | 0 | 0 | 0 | 0 |

Figure 10: The Feature Ranking Matrix $R^2$ with the top two features from the top 12 common features.

|    | 0 | 1 | 2 | 3 | 4 | 5 | 6 | 7 | 8 | 9 | 10 | 11 | 12 | 13 | 14 | 15 |
|----|---|---|---|---|---|---|---|---|---|---|----|----|----|----|----|----|
| 0  | 0 | 0 | 0 | 0 | 0 | 0 | 0 | 0 | 0 | 0 | 0 | 0 | 0 | 0 | 0 | 0 |
| 1  | 0 | 0 | 0 | 0 | 0 | 0 | 0 | 0 | 0 | 0 | 0 | 0 | 0 | 0 | 0 | 0 |
| 2  | 0 | 0 | 0 | 0 | 0 | 0 | 0 | 0 | 0 | 0 | 0 | 0 | 0 | 0 | 0 | 0 |
| 3  | 0 | 0 | 0 | 0 | 0 | 0 | 0 | 0 | 0 | 0 | 0 | 0 | 0 | 0 | 0 | 0 |
| 4  | 0 | 0 | 0 | 0 | 0 | 0 | 0 | 0 | 0 | 0 | 0 | 0 | 0 | 0 | 0 | 0 |
| 5  | 0 | 0 | 0 | 0 | 0 | 0 | 0 | 0 | 0 | 0 | 0 | 0 | 0 | 0 | 0 | 0 |
| 6  | 0 | 0 | 0 | 0 | 0 | 0 | 0 | 0 | 0 | 0 | 0 | 0 | 0 | 0 | 0 | 0 |
| 7  | 0 | 0 | 0 | 0 | 0 | 0 | 0 | 0 | 0 | 0 | 0 | 0 | 0 | 0 | 0 | 0 |
| 8  | 0 | 0 | 0 | 0 | 0 | 0 | 0 | 0 | 0 | 0 | 0 | 0 | 0 | 0 | 0 | 0 |
| 9  | 0 | 0 | 0 | 0 | 0 | 0 | 0 | 0 | 0 | 0 | 0 | 0 | 0 | 0 | 0 | 0 |
| 10 | 0 | 0 | 0 | 0 | 0 | 0 | 0 | 0 | 0 | 0 | 0 | 0 | 0 | 0 | 0 | 0 |
| 11 | 0 | 0 | 0 | 0 | 0 | 0 | 0 | 7 | 0 | 0 | 0 | 0 | 0 | 0 | 0 | 0 |
| 12 | 0 | 0 | 0 | 0 | 0 | 0 | 0 | 0 | 0 | 0 | 0 | 0 | 0 | 0 | 0 | 0 |
| 13 | 0 | 0 | 0 | 0 | 0 | 0 | 0 | 0 | 0 | 0 | 0 | 0 | 0 | 0 | 0 | 0 |
| 14 | 0 | 0 | 0 | 0 | 0 | 0 | 0 | 0 | 0 | 0 | 0 | 0 | 0 | 0 | 0 | 0 |
| 15 | 0 | 0 | 0 | 0 | 0 | 0 | 0 | 0 | 0 | 0 | 0 | 0 | 0 | 0 | 0 | 0 |

Figure 11: The Feature Ranking Matrix $R^3$ with one top feature from the top 12 common features.

