# OpenReview forum: "Patch Ranking Map: Explaining Relations among Top-Ranked Patches, Top-Ranked Features and Decisions of Convolutional Neural Networks for Image Classification"
_ICLR.cc/2024/Conference — Submitted to ICLR 2024_

### Official Review · Reviewer_1SjE · 2023-10-30

**Soundness:** 3 good
**Presentation:** 2 fair
**Contribution:** 2 fair
**Rating:** 5
**Confidence:** 4

**Summary:**

The manuscript introduces an explainable convolutional neural network (CNN)-based method for medical image classification. Specifically, it first uses a multi-phase feature selection algorithm to select important features from a pre-trained CNN model, and then adopts a patch ranking algorithm to identify informative patches in the input image. The method is evaluated on an Alzheimer’s magnetic resonance imaging (MRI) dataset to demonstrate its effectiveness.

**Strengths:**

1. The manuscript presents a simple method to identify important regions/patches in input images for CNN predictions and thus helps improve model explainability.

2. The proposed method can produce higher image classification accuracy with a smaller model size.

**Weaknesses:**

1. The focus of this submission is to improve CNN model explainability, but this is not well verified in the experiments. For instance, it is not clear how the most informative patches “at the 7th and 12th patch rows and at the 8th and 9th patch columns” identified by the method is related to the clinical interpretation. Are those patches indeed the most discriminative from a clinical perspective? In addition, the quantitative evaluation of how the top-ranked image patches contribute to the CNN prediction is also not clear.

2. The presentation is difficult to follow and needs to be significantly improved. (1) Some key technical concepts or details are not well explained. For example, the motivation of patch ranking score definition in Algorithm 2 is not clear, and no intuitive explanation is given in the manuscript. In addition, the introduce of \omega_k is not well motivated. (2) The notations are inconsistent. For instance, do the K’s in Algorithm 1 and Algorithm 2 have the same value? In Section 2.1, m = n x H x W and represents the number of the features, but m = max(a_{ij}) in Section 4. Additionally, no definition for Q_S (Section 2.1) is provided. These ambiguous notations are very confusing. (3) Figure 1 does not clearly show the workflow of the proposed method in the training or testing stage. It would be helpful to provide two separate figures for model training and testing respectively.

3. The method is tested on a single CNN backbone, ResNet50, which produces features with each corresponding to 8 x 8 patches in the input image. It is not clear if the method is applicable to other CNN architectures that extract features with each corresponding to different-sized image patches. This is not explored or tested in the experiments.

4. The proposed method is evaluated on a single image dataset and disease, so the generalization to other diseases is unknown.

5. The dataset used in the experiments is highly imbalanced. The number of moderate demented cases (64 subjects) is much smaller than that of non-demented (3200 subjects) or very mild demented (2240 subjects) cases. Is it not clear what loss is used to handle the data imbalance. In addition, the model classification performance is evaluated using only classification accuracy, which is not an appropriate choice for imbalanced data.

**Questions:**

In addition to the comments in the Weaknesses above, there are some others to be clarified:

1. It seems that the proposed method requires the input images to be well registered so that all images are in the same coordinate system. What if some input images shift to a direction (horizontal or vertical) by a few pixels? Will this affect the identification of the important image patches for different input images that are not in the same coordinate system or space?

2. In Section 1 of Page 2, regarding the sentence “current methods such as CAM-methods based heatmaps do not deeply analyze a relationship among ranked image patches, top features, top feature maps, and a decision.” What does “deeply analyze a relationship” mean?

---

### Official Review · Reviewer_xK6b · 2023-10-31

**Soundness:** 3 good
**Presentation:** 2 fair
**Contribution:** 2 fair
**Rating:** 3
**Confidence:** 3

**Summary:**

This paper describes a Patch Ranking Map (PRM), a learnable set of image patches that have important associations with a CNN’s decisions. A feature accumulation matrix and feature ranking matrix are used to rank image patches in the PRM for purposes of explainability. The PRM was shown to consistently select brain regions associated with Alzheimer’s disease from MRI images, and the selection of top features was shown to enable more efficient models that improve test accuracy, while reducing model size (Table 1).

**Strengths:**

-	Novel framework for systematic ranking of image patches, from feature maps
-	Demonstrates possibility for improved classification performance by eliminating relatively uninformative features, which is relevant in medical domains with limited data

**Weaknesses:**

-	PRM method appears relevant only for tasks where inputs are strictly registered according to some atlas, such that corresponding regions are always in the same spatial position
-	Minimal details on the MLP classifying the extracted selected top features
-	Actual contribution to explainability above existing heatmap methods unclear

**Questions:**

1. The H x W dimensions of the image patches might be justified. Are they due to the nature of the MRI task? In any case, would 1x1 patches (i.e. H = W = 1) equivalent to pixel-level heatmaps be feasible?
2. The lack of details about the MLP model applied to the selected top features is critical, since it would appear to significantly affect ultimate model performance.
3. For the model size values listed in Table 1, it might be clarified as to whether this includes the ResNet50 feature extractor, since that would be a part of the ResNet50-FS model during inference. Moreover, again, the lack of details about the MLP makes it difficult to comprehend the tradeoff with respect to selected top features, and model size.
4. The method of combining the different FS methods (via the feature matrices) might be further substantiated, especially with respect to the weights of the weighted average functions mentioned in Section 4. How were the weights determined?

---

### Official Review · Reviewer_PHrZ · 2023-11-01

**Soundness:** 2 fair
**Presentation:** 1 poor
**Contribution:** 2 fair
**Rating:** 1
**Confidence:** 3

**Summary:**

The paper proposed a CNN architecture that incorporate feature selection method, in order to provide more explainability to the model decision.

**Strengths:**

The paper aims to tackle an important problem, this is especially important in medical applications.

**Weaknesses:**

1. The novelty is limited, the proposed model mostly involves straight application existing feature selection methods.
2. The evaluation is limited (only on one dataset)
3. Presentations could also be improved.
4. Incomplete study, the relationship between the top selected patches and the disease is not yet established

**Questions:**

NA

---

### Official Review · Reviewer_tnrN · 2023-11-01

**Soundness:** 1 poor
**Presentation:** 2 fair
**Contribution:** 1 poor
**Rating:** 1
**Confidence:** 4

**Summary:**

A method that incorporates feature selection algorithms into a CNN image classifier is presented that results in a ranking of image patches in input space. This ranking reflects the relevance of the patches to the classifier, and thus provides a way to visualize which part of the images are important to the classification task. Classification of brain MRI images from an Alzheimer’s dataset is used in an experiment and a preliminary result is given but without relating it to the relevant medical literature.

**Strengths:**

The attempt to visualize which parts of the brain are implicated in classifying disease can be of relevance in some applications. The authors have used existing feature selection methods to do this in a novel way.

**Weaknesses:**

The approach lacks clear theoretical motivation, and is not compared with existing methods for obtaining sparsity in neural networks or obtaining visual “explanations”. The result highlights certain locations/patches as being most relevant, but this preliminary work has not yet ascertained whether this result makes sense to clinicians/neurologists or agrees with what is known from the medical literature. There are various aspects of the method and experiments which need further development.

Algorithm 1 applies m feature selection methods sequentially, each reducing the number of features by a prespecified amount, finally reaching K features. This Algorithm box needs to be accompanied by some more careful motivation and explanation. Why is this a good idea? Does it matter which methods are used and on what order, and how might that be decided? How are the numbers of features at each stage to be determined in advance?

Algorithm 2 effectively just states “select the K patches having the largest values of theta = f (a,r) for some function f”. This doesn’t need its own algorithm box.  The motivation for the choice of f() given on p4 was not clear to me. It is stated without giving the theoretical motivation.
The first paragraph of 5.1 gives a part of the method, in which a fine-tuned network is combined with feature selection and a newly trained MLP. This needs to be described more carefully in Section 4.

Details are missing from the experiment description which states that “top features are selected by the 5-phase FS algorithm”. Which FS methods make up the 5-phase algorithm, in which order were these applied, and how were the parameters omega_0 and s determined?

Table 1 compares the methods with a method that uses all the features. Can the authors please clarify whether the latter uses the extra MLP model, in common with the proposed method? If not, is the comparison a fair one?

Table 3 reports results using 7 partial feature sets, 6 features, from 7 100-feature sets. How/why were these numbers (7, 6, 7, 100) chosen? Were they determined in advance of the experiment, or are they the result of some hyperparameter search which has not been described?

Some sections need reorganized or rewritten. Section 1 contains unnecessary repetition and use of terms such as “ranked image patches”, “top features”, “top feature maps” without first really saying what they mean (top of what?). Section 2 uses notation that could certainly be simplified, and rather than start with a list of definitions would be better to start with the overview which is currently at the start of Section 4. (I think k in Defn. 6 should be K).

**Questions:**

See "weaknesses" section above.

---

### Meta-Review · Area_Chair_JL7A · 2023-12-06

**Metareview:**

This paper presents a method for explainable AI for classification. Although the title seems to cover a broad application of classifications, the experimental validation is limited to brain MRI. The paper has multiple weaknesses including the motivation (R1), the technical novelty (R1, R2, R4) and also experimental validations (R1-R4). Overall, the work is clearly under the standard of this conference.

**Justification For Why Not Higher Score:**

The paper has multiple flaws to consider it for acceptance.

**Justification For Why Not Lower Score:**

NA

---

### Decision · Program_Chairs · 2024-01-16

Reject